## COMMENT

# Resisting disinfectants

Harrie F. G. van Dijk [1✉], Henri A. Verbrugh[2] & Ad hoc advisory committee on disinfectants of the Health Council of the Netherlands*

Although often overlooked, the use of disinfectants can lead to antimicrobial resistance and this may exacerbate resistance to antibiotics. Here, we explain why all antimicrobial agents, including disinfectants, should be used prudently in a way that is guided by evidence.

Disinfectants are antimicrobial products that incorporate one or more active substances, such as chlorine, iodine, alcohols, hydrogen peroxide, silver, chlorhexidine, triclosan and quaternary ammonium compounds. They are indispensable in human and veterinary health care, the food industry and water treatment for the prevention of infections and intoxications. Presently, their use in public and private domains is increasing: the current COVID-19 pandemic has boosted[1] an already ongoing trend of an increasing array of consumer products that contain disinfectants[2]. Whereas resistance to antibiotics is regarded as a major health threat, resistance to disinfectants is receiving little attention from practitioners in human and veterinary health care and in food production, and from administrators and authorities. One potential reason is the lack of a broadly accepted definition of resistance to disinfectants[3]. Another is that the reductions in susceptibility to disinfectants commonly observed in settings of frequent use are mostly modest. Minimum concentrations of disinfectants needed to arrest the growth of strains (minimum inhibitory concentration (MIC)) or to kill strains (minimum bactericidal concentration (MBC)) isolated from such places are normally less than ten times higher than the MIC or MBC of strains from settings where disinfectants are hardly used. As in-use concentrations of disinfectants are still considerably higher, the relevance of these moderate increases in MIC or MBC is disputed[3]. We fear to be trapped in a vicious circle of the presumed insignificance of resistance to disinfectants and the lack of attention for it. While the term 'tolerance' for these lower levels of resistance to disinfectants is often used[4], we think that this term downplays the importance of the phenomenon. In line with EUCAST[5], we advocate to use the term 'tolerance' only in cases where the MBC of a strain is much increased, such that the strain is not readily killed anymore, while its MIC remains unchanged. In a microbiological sense, the term 'resistance' is used to denote any reduction in susceptibility demonstrated phenotypically by increases in MIC or MBC.

The emergence of resistance is the inevitable consequence of all use of disinfectants, rather than just improper use. Generally, a heterogeneous community of bacteria is present at the application site, consisting of species and strains that are more and less susceptible to the disinfectant. Individual cells may be resistant enough to survive a disinfection procedure. In addition, microorganisms may reside in dirt, in nooks and crannies and in biofilms, where disinfectants cannot reach easily. In these places, and at the margins of the disinfected area, microorganisms are exposed to lower disinfectant concentrations enabling less susceptible strains to survive. Disinfectants end up in surface waters or soil via sewer lines and fertilizing manure[6]. Dilution and degradation result in environmental levels that are much lower than those used at the point of application.

Exposure to sub-MIC concentrations of disinfectants can trigger stress responses in bacteria that induce temporary, adaptive changes in the composition and permeability of their cell envelopes[7] or in the activity of their efflux pumps[8]. They also trigger an increase in the frequency

[1] Health Council of The Netherlands, P/O Box 16052, 2500 BB The Hague, The Netherlands. [2] Erasmus University Medical Centre, Rotterdam, The Netherlands. *A list of authors and their affiliations appears at the end of the paper. ✉email: hfg.van.dijk@gr.nl

**Table 1 Outbreaks of bacterial infections in hospitals where the susceptibility of the causative pathogen for the disinfectant used was investigated.**

| Bacterial species | Disinfectant | Growth inhibition experiments | Time-killing and survival tests | Remarks | Reference |
|---|---|---|---|---|---|
| Klebsiella pneumoniae | Peracetic acid-based solution, strength not specified (normal 0.2%) | Not done | >5 log reduction in 5 min | Tested according to European standard EN 13727 test | Endoscopy 2010;42:895 |
| Salmonella kedougou | 1% Savlon (cetrimide + chlorhexidine in a 10:1 ratio) | Not done | Strain remained viable on artificially contaminated and then disinfected gastroscope | | Lancet 1982;2:864 |
| Mycobacterium tuberculosis | Iodophor (povidone-iodine) solutions | Not done | survived 10->30 min in (un)diluted iodophors | Solutions were 2 parts iodophor, 1 part 70% ethyl alcohol and 1 part sterile water | Am Rev Respir Dis 1983;127:97 |
| Mycobacterium fortuitum | 2% glutaraldehyde | Not done | Survived >30 min on isolated, artificially contaminated suction valves of bronchoscope | Viable mycobacteria on valves after disinfection in a commercial endoscope washer | J Infect Dis 1989;159:954 |
| Pseudomonas sp. (3 isolates) | 0.05% aqueous chlorhexidine and 1% Savlon (cetrimide + chlorhexidine in 10:1 ratio) | 1 strain grew 100,000-fold in 0.05% chlorhexidine in 3 weeks | All 3 strains survived in 0.05% chlorhexidine, 1 in 0.1% solution | | Br Med J 1967;2:153 |
| Serratia marcescens | 2% aqueous chlorhexidine | Outbreak strain's minimum inhibitory concentration 1024 mg/L (vs. 16 mg/L for control strains) | Survived for up to 27 months in 2% aqueous chlorhexidine | By EM cells had altered cytoplasms, while kept in 2% chlorhexidine | Appl Environ Microbiol 1981;42:1093 |
| Pseudomonas cepacia | Aqueous chlorhexidine (0.05–0.2%) | Outbreak strain growth inhibited at ±200 mg/L (versus <50 mg/L for control strains of this species) | Not done | Growth inhibition was studied in aqueous chlorhexidine, not in broth growth medium | Am J Med 1982;73:183 |
| Alcaligenes xylosoxidans | Aqueous chlorhexidine (at 0.6 g/L) | Not done | >5 and 2 g/L required to kill strain within 5 and 60 min, respectively (0.025 and 0.005 g/L for Pseudomonas aeruginosa ATCC reference strain) | Referred to European standard EN 1040 test | Eur J Clin Microbiol Infect Dis 1998;17:724 |
| Pseudomonas multivorans | 1:30 Savlon (cetrimide + chlorhexidine in 10:1 ratio) | Minimum inhibitory concentration Savlon was 1:320, but variants requiring higher concentrations were selected | Selected variants survived in 1:30 Savlon | 4 weeks growth in 1:1 mixture of 1:30 Savlon and 1% Peptone, continued growth when subcultured in 1:30 Savlon alone | Lancet 1970;1:1188 |

**Table 1 (continued)**

| Bacterial species | Disinfectant | Growth inhibition experiments | Time-killing and survival tests | Remarks | Reference |
|---|---|---|---|---|---|
| Pseudomonas sp. | 0.1% aqueous benzalkonium chloride | Not done | At 0.1% benzalkonium chloride killing required 15 min, at 0.01% all strains survived >24 h | Bacteria survived in 0.4% benzalkonium chloride diluted in growth medium (TSB) | Am J Med Sci 1958;235:621 |
| Pseudomonas-Achromobacteriaceae sp. | 0.1% aqueous benzalkonium chloride | Not done | Bacteria survived >24 h in 0.1% benzalkonium solution with cotton pledgets | Cotton likely absorbed some benzalkonium chloride but this species is relatively resistant | JAMA 1961;177:708 |
| Serratia marcescens | 0.13% aqueous benzalkonium chloride | Not done | Outbreak strain survived 0.13% benzalkonium chloride for 10–60 min, control strains did not | Found cross-resistance to picolinium chloride, used as a preservative in medicines | J Clin Microbiol 1987;25:1014 and 1019 |
| Mycobacterium abscessus | 0.13% benzalkonium chloride | By disk diffusion 9–11 mm growth inhibition found at 0.13% but not at 0.013% benzalkonium chloride | Outbreak strain survived 0.01% benzalkonium chloride, ATCC type strain survived in 0.13% | | Clin Infect Dis 2003;36:954 |

of gene mutations[9] and stimulate horizontal gene transfer (HGT)[10], which may result in the acquisition of new resistance mechanisms. Low-level exposure to antimicrobial substances selects mutants with low fitness costs that persist once the substance has dissipated[11]. Moreover, resistant bacteria may mitigate fitness costs by acquiring additional mutations or genes[12]. Competition experiments in the laboratory have shown that exposure to concentrations of antimicrobials far below the MIC enables bacteria with a reduced antimicrobial susceptibility to gradually outcompete their more susceptible kin[13]. Thus, due to sub-MIC concentrations of disinfectants, selection of less susceptible bacteria may occur even in the environment[14].

The fact that microbial resistance to in-use concentrations of disinfectants has only sporadically been observed in practice offers us little reassurance. Outbreaks of healthcare-associated infections connected with the use of disinfectants are regularly described in the scientific literature[15–17]. Often, they result from failing disinfection procedures for reusable medical devices such as endoscopes, or from the use of disinfectant solutions, which are contaminated themselves. Investigations into such incidents concentrate on tracing sources and transmission routes of the bacteria involved, and on finding errors in the disinfection procedures followed. However, to what extent does resistance of the pathogen involved to the disinfectant used play a role in these outbreaks? To find an answer to this question, we conducted an analysis as part of the preparation of an advisory report of the Health Council of the Netherlands[18]. We scanned all publications referenced in three review papers[15–17] for information on the pathogen's susceptibility to the disinfectant involved. After removing duplicates, 138 publications remained. Due to their age, 11 articles were not accessible via the Erasmus MC E-library. In 6 publications, the pathogen was not isolated. Mostly, this concerned procedural papers and questionnaires. In 108 publications, the pathogen's susceptibility to the disinfectant used was not determined, although its susceptibility to antibiotics was investigated in many cases. In 13 publications, the pathogen's susceptibility to the disinfectant was assessed (Table 1). In 12 cases, the pathogen turned out to be highly resistant to the disinfectant applied.

In conclusion, in almost 90% of the outbreaks in which a pathogen was isolated, its susceptibility to the disinfectant used was not determined. When it was assessed, in most cases the pathogen turned out to be highly resistant to the disinfectant. Therefore, resistance to disinfectants seems to play an important role in incidents involving disinfection failure. However, investigators fail to consider it as a possible (co-)determinant. Clinical laboratories' inexperience in determining susceptibility to disinfectants may be an additional explanation.

Moderate levels of resistance (up to ten times higher MIC values) might be relevant as well, as is illustrated by the persistence of *Listeria* in the food industry[19]. The fact that concentrations of disinfectants used in practice are much higher than the elevated MIC values commonly observed does not guarantee a successful disinfection. Disinfectants are often required to do their work within minutes. Contact between the disinfectant and microorganisms at the site of application is often not optimal. Bacteria may reside in places that are difficult to reach by the disinfectant. Also, volatile disinfectants may dissipate too rapidly, while others may be inactivated by organic material. Moreover, microorganisms may upregulate their repair mechanisms. Each reduction in bacterial susceptibility, however small, may further compromise the effectiveness of disinfection. It is by combining several strategies that microorganisms succeed in surviving disinfections.

Mechanisms that reduce a microorganism's susceptibility to a disinfectant may also diminish its susceptibility to other disinfectants and antibiotics, a phenomenon called cross-resistance[20]. Antimicrobial resistance genes tend to be genetically linked, by co-residing on plasmids or integrative and conjugative elements, and thus transferred together, paving the way for co-selection[21]. Exposure to disinfectants may stimulate biofilm formation[22] or drive bacteria into a metabolically inactive state[23], rendering infections more difficult to cure with antibiotics. It may also promote HGT of antibiotic resistance genes[10]. There is evidence that the use of quaternary ammonium compounds and sulphonamides since the 1930s has facilitated the spread of class 1 integrons and, thus, the evolution of antibiotic resistance in clinically relevant bacteria[24]. In the laboratory, repeated exposure to disinfectants has been shown to result in the selection of bacteria with reduced susceptibility to antibiotics[21]. While the degree to which disinfectant use contributes to the emergence of antibiotic resistance in practice remains unclear, the need for more data on this topic is evident.

We recommend that governments assure that in various relevant sectors professionals, in close cooperation with administrators, develop and implement policies to promote the prudent use of disinfectants. In professional sectors, disinfectants should be applied according to evidence-based guidelines specifying when their use has a proven added value in preventing or controlling infection or damage, e.g., food spoilage. Private individuals should only use chemical disinfectants when prescribed by a medical doctor or other qualified experts. In line with international recommendations[25], health, cosmetic and aesthetic objectives should be pursued without the use of chemical disinfectants whenever possible. In many cases, regular and thorough cleaning with water and a detergent may suffice.

We further recommend setting up a robust surveillance system for monitoring the consumption of disinfectants and the development of resistance. Initially, this may be limited to human and veterinary health care sectors. There, the risk of resistance development seems to be greatest, and test facilities are amply available. If the results suggest a need, surveillance may be extended to encompass other sectors. It may be wise to start at a national level, but to strive for international cooperation. The antibiotic resistance surveillance at the European level, which is coordinated by the European Centre for Disease Prevention and Control, may serve as an example. We expect that efforts to halt or even curb antibiotic resistance will benefit from also paying due attention to disinfectant resistance.

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

## Acknowledgements

We would like to acknowledge that the main authors were the chairperson (HAV) and scientific secretary (HFGvD) of the ad hoc advisory committee of the Health Council of the Netherlands that produced the advisory report Careful use of disinfectants (https://www.healthcouncil.nl/documents/advisory-reports/2016/12/21/careful-use-of-disinfectants). The other authors served on the same committee as member or advisor. Please note that the authors have written this article in a personal capacity, and that the article, therefore, does not necessarily reflect the opinions of the Health Council of the Netherlands.

## Author contributions

H.F.G.v.D. and H.A.V. conceived the idea for this comment, H.F.G.v.D. wrote the initial text and managed its subsequent production and submission to *Communications Medicine*, whilst the other authors were actively involved in contributing to and shaping the sequential iterations of the text and approved its final version.

## Competing interests

The authors declare no competing interests.

## Additional information

## Ad hoc advisory committee on disinfectants of the Health Council of the Netherlands

Tjakko Abee[3], Jan Willem Andriessen[4], Harrie F. G. van Dijk[1], Benno H. ter Kuile[5], Dik J. Mevius[6], Mark H. M. M. Montforts[7], Willem van Schaik[8], Heike Schmitt[9], Hauke Smidt[3], Jan-Willem Veening[10], Henri A. Verbrugh[2] & Andreas Voss[11]

[3]Wageningen University & Research, Wageningen, The Netherlands. [4]Board for the Authorisation of Plant Protection Products and Biocides, Ede, The Netherlands. [5]University of Amsterdam, Amsterdam, The Netherlands. [6]Utrecht University, Utrecht, The Netherlands. [7]National Institute for Public Health and the Environment, Bilthoven, The Netherlands. [8]University of Birmingham, Birmingham, UK. [9]Institute for Risk Assessment Sciences, Utrecht University, Utrecht, The Netherlands. [10]University of Lausanne, Lausanne, Switzerland. [11]Radboud University Medical Center, Nijmegen, The Netherlands.

