## [Peer Review File · Communications Medicine]

Reviewers' comments:

Reviewer #1 (Remarks to the Author):

Overall comments:

The short commentary by authors Dr. van Dijk and Verbrugh on disinfectant resistance is a concise and timely call to arms for those studying antimicrobial resistance. Although the authors do not offer solutions to the problem or make suggestions on how to improve disinfectant resistance surveillance, usage, study or its definitions, this brief literature review (and its table summarizing past studies that have actually explored disinfectant outbreak susceptibility values) is a useful addition to currently available disinfectant reviews. Given that the main attributes of the study focus on a literature review of disinfectant outbreaks in Table 1, it would be useful for readers interested in this subject to add MIC values (if available) as an additional column to table 1. In the written text, the authors need to clearly define what disinfectant resistance is as they perceive it, as there are currently no defined CLSI or EUCAST breakpoints or recognized MIC values for any of the disinfectants listed in their commentary. The authors often refer to "moderate" or "low" resistance for disinfectants but it was unclear what these terms mean without a value such as MIC as a frame of reference. Adding these will help strengthen this brief and useful communication. There are a number of specific comments to help improve some statements and their clarity/wording below.

Specific comments:

In the summary consider replacing "often ignored" with "over-looked" or explain who is ignoring their study, since there are active research groups that are exploring disinfectant resistance mechanisms.

In paragraph 1 line 1: Wording - Please consider replacing "...based on active ingredients.." with "that incorporate active ingredients.." since some disinfectants are on their own chlorine and iodine etc.

Paragraph 1 line 3: "They are indispensable in..." Please explain what they are indispensable for.

Paragraph 1 line 4. "Presently their use in public and private domains is increasing." This is a sentence fragment. Please clarify why disinfectant usage is increasing (is this pandemic related or just due to improved hygienic practices?) and for what compounds? This will help educate the reader.

Paragraph 1 line 6: please omit "probably" from the statement "One reason probably is the lack of a ..."

Paragraph 1, line 9. Please explain what is considered a "modest" MIC reduction in this statement. It is essential to define this more clearly so a reader is aware of what the authors define modest as based on an MICs (in fold changes) this varies considerably in publications and is discussed further in the paragraph.

Paragraph 1, line 12. Do the authors mean "gradual" or are they referring to "additive" resistance in this statement? Please briefly clarify what is meant here with gradual lower and higher resistance levels.

Paragraph 1 lines 12-13. Is there evidence to support the fear that disinfectants and therapeutic antimicrobials perpetuate overall resistance?

Paragraph 2 line 2. "Rather, it is the inevitable consequence of all use." Is "all" referring to any antimicrobial here? Also this is a sentence fragment; please consider using a semi-colon and merge it to the end of the first statement or rephrase.

Paragraph 2 line 3. Please add "disinfectant" to the start of "...at the application site,..." . Also in this statement "...more and less susceptible.." is this referring to antimicrobial or disinfectant susceptible species/ strains? Please clarify.

Paragraph 2 line 4. "Individual cells may be resistant enough to survive a disinfection procedure." What is meant by "individual" are planktonic cells being referred to here (as the next sentence discusses biofilms)? Most studies examining bacterial cell killing when grown planktonically are more susceptible to disinfectants not more resistant. Please rephrase and consider including some cited studies to support the statement.

Paragraph 2 line 5. "...microorganisms may shelter in dirt..." What is meant by shelter? Technically this is an anthropomorphism. Perhaps "accumulate" or "enrich" is a more appropriate term?

Paragraph 2 line 6. Please add "disinfectant" in between "lower" and "concentrations" in this statement.

Paragraph 3 line 4. Wording – "...which may result in acquired resistance." This seems to imply an increase in acquired resistance mechanisms?

Paragraph 3 lines 7-8. Please place references before the period so the reader knows what statement they are referring to this is inconsistently formatted in other statements in the article.

Paragraph 3 line 7 "...selection of less susceptible bacteria may occur even in the environment." Please clarify that environments have low or sub-inhibitory levels of disinfectants. It would be good to include a statement regarding worldwide disinfectant pollution to strengthen this argument.

Paragraph 3 line 9. "Moreover, resistant bacteria..." what resistance is being referred to here? Disinfectant or antimicrobial or both? It's unclear as written.

Paragraph 4 line 2. "...observed in practice offers little reassurance." Who are the stakeholders referred to in this statement? Please clarify.

Paragraph 4 line 4-5. "...routes of incriminated microorganisms,.." what is meant by this term? Is contaminating meant here?

Paragraph 5 lines 2-3. "Disinfectants are required to do their work within minutes." Technically disinfectant mechanisms of action also require minute timeframes, perhaps this can be stated?.

Paragraph 5 line 4. "...application is often not optimal." Please give brief examples of how disinfectant contact is not optimal.

Paragraph 5 line 6. Technically bacteria do not “hide in dirt” (this is an anthropomorphism), but the authors argument that the presence of obscuring materials like dirt would impose more barriers and dilute the disinfectant as it reaches and acts on bacteria.

Paragraph 6 line 2. “Antimicrobial resistance genes tend to be genetically linked...” What is meant by this statement? This is not necessarily true and quite context dependent (eg. Are the linked genes part of an intrinsic resistance vs acquired resistance system, and are linked genes controlled by antimicrobial triggered regulons such as mar-sox-rob system?). It is important to clarify the meaning here to understand what is meant by co-selection and in what form (on plasmids, mobile elements chromosomes)?.

Paragraph 6 lines 5-6. “It may also promote HGT of antibiotic resistance genes.” Unclear antecedent (what is “it”) and sentence fragment.

Paragraph 6 line 11. “..., the need for better data is evident.” Is better data meant here or more experimental analyses of disinfectant resistance and its associated mechanisms?

Paragraph 7 lines 1-2. “...,we advocate a prudent approach to all antimicrobials.” What is meant by a prudent approach? Unclear.

Paragraph 7 lines 2-3. Can the authors cite supporting literature to support the statement made regarding microbiome toxicity and disinfectant exposure?

Comments related to content in boxed section entitled “Outbreaks of bacterial infections in hospitals related to disinfectants”

What is meant by “redoubling” in line 10 of boxed text for the statement “After de-doubling, this resulted in 138...”.

In bullet point 4 of the boxed section, “The pathogen was tested for susceptibility to the disinfectant in 13 publications (see Table 1). In most cases, the pathogen was found to be highly resistant to the disinfectant used.” Please provide the total number of articles where the pathogen was found to be resistant to the disinfectant. Again how are the authors defining disinfectant resistance (2-fold MIC, 4 fold MIC changes >4 fold MIC) when reviewing these studies?

Boxed In summary lines 2-3: “When such a determination was performed, the pathogen was found to be highly resistant to the disinfectant in question in most cases.” In most cases undermines that statements made prior. Please give a value.

For Box Conclusion line 1: “Resistance to disinfectants likely does play an important role” please remove the word “does” and pluralize “play”.

Table 1. Please consider bolding the font and using upper case headings for column titles and adding row lines at minimum to help keep table cell content ordered on each row (some parts of the table are hard to follow). Also please consider using footnotes to abbreviate some repetitive words like chlorhexidine (etc) to reduce text so findings are clearer and font size can be increased. Please define abbreviate terms such as EM. Please use a footnote to reference time killing and survival experiment methods that are referred to in Table column header 4. Please format references

consistently and by number in this table.

Reviewer #2 (Remarks to the Author):

The authors describe a relevant topic. Based on my knowledge of the literature I find a few major concerns that should be addressed before a final recommendation may be given.

1. I strongly suggest to use the current terminology used in the European Union. Disinfectants are biocidal products based on different biocidal active substances. This is important because disinfectants may be based on two or more different biocidal active substances. Increased MIC values indicating cellular tolerance are typically determined using a single biocidal active substance although it may be possible to determine an MIC value using the entire formulation. This aspect should be considered in the entire manuscript.
2. I agree that there is a lack of international definition to determine “resistance”. But I am also not clear about the use of the terms “tolerance” and “resistance” in the manuscript. My understanding is that tolerance describes any type of elevated MIC or MBC values compared to comparable isolates or strains of a species. The term “resistance” may indicate an epidemiological resistance as suggested by Morrissey et al. (10.1371/journal.pone.0086669) or a clinical resistance describing an isolate with a lower log reduction compared to type of recommended application (10.1016/s1473-3099(03)00833-8). This aspect should also be considered in the entire manuscript.
3. “sub-lethal concentrations”: please include sub-inhibitory concentration because they have been studied in numerous publications with an adaptive cellular response.
4. “Exposure to very low concentrations”: please be more specific (see 3.).
5. Page 3, paragraph 3: please consider to add volatility of an important factor for a possible adaptation (duration of exposure) and stability of the chemical (autocatalytic substances; also duration of exposure and concentration during exposure).
6. Page 3, last paragraph: cross-tolerance to other biocidal active agents also occurs and would be worth to mention with one or two examples.

Response to referees

Point raised by reviewers	Response by authors
Reviewer 1	
Summary: consider replacing "often ignored" with "overlooked"	"Although often overlooked, the use of ..."
Par. 1 line 1: consider replacing "...based on active ingredients.." with "that incorporate active ingredients.."	"Disinfectants are antimicrobial products that incorporate one or more active substances such as ..."
Par. 1 line 3: "They are indispensable in..." Explain what they are indispensable for.	"They are indispensable in human and veterinary health care, the food industry and water treatment for the prevention of infections and intoxications."
Par. 1 line 4. "Presently their use in public and private domains is increasing." Clarify why disinfectant usage is increasing (is this pandemic related or just due to improved hygienic practices?) and for what compounds?	"The current COVID-19 pandemic has boosted an already ongoing trend of an increasing array of consumer products that contain disinfectants." Regarding the ongoing trend, we made reference to a report of the Swedish Chemicals Agency. We think it is too much detail to be more specific on the compounds.
Par. 1 line 6: omit "probably" from the statement "One reason probably is the lack of a ..."	"One reason is the lack of a broadly accepted definition of resistance to disinfectants."
Par. 1, line 9. Explain what is considered a "modest" MIC reduction in this statement.	"Minimum concentrations needed to arrest the growth of strains (Minimum Inhibitory Concentration, MIC) or to kill strains (Minimum Bactericidal Concentration, MBC) isolated from such places are normally less than 10 times higher than the MIC or MBC of strains from unexposed settings."
Par. 1, line 12. Do the authors mean "gradual" or are they referring to "additive" resistance in this statement? Clarify what is meant here with gradual lower and higher resistance levels	We deleted this statement. It became superfluous after we more clearly defined what we mean by 'resistance' and by referring to the definition in the microbiological sense of EUCAST.
Par. 1 lines 12-13. Is there evidence to support the fear that disinfectants and therapeutic antimicrobials perpetuate overall resistance?	We think that our statement was misunderstood. What we mean to say is that the lack of attention to disinfectant resistance is perpetuating its presumed insignificance and vice versa.
Par. 2 line 2. "Rather, it is the inevitable consequence of all use." Is "all" referring to any antimicrobial here? Also this is a sentence fragment; please consider using a semi-colon and merge it to the end of the first statement or rephrase.	"A misconception is that resistance emergence is primarily the result of applying antimicrobials improperly; rather, it is the inevitable consequence of all use." "All" is referring to both improper and proper uses.
Par. 2 line 3. Add "disinfectant" to the start of "...at the application site,....". Also in this statement "...more and less susceptible.." is this referring to antimicrobial or disinfectant susceptible species/strains?	"Generally, a heterogeneous community of bacteria is present at the application site, consisting of species and strains that are more and less susceptible to the disinfectant."
Par. 2 line 4. "Individual cells may be resistant enough to survive a disinfection procedure." What is meant by "individual" are planktonic cells being referred to here (as the next sentence discusses biofilms)? Most studies examining bacterial cell killing when grown planktonically are more susceptible to disinfectants not more resistant. Please rephrase and consider including some cited studies to support the statement.	Our statement may be misunderstood. What we mean to say is that within a population of bacteria (whether it is living planktonic or in a biofilm), there are inter-individual differences in susceptibility to a disinfectant.
Par. 2 line 5. "...microorganisms may shelter in dirt..." What is meant by shelter? Technically this is an anthropomorphism. Perhaps "accumulate" or "enrich" is a more appropriate term?	"Additionally, microorganisms may reside in dirt, in nooks and crannies, and ..."
Par. 2 line 6. Please add "disinfectant" in between "lower" and "concentrations" in this statement.	"There, and at the margins of the disinfected area, they are exposed to lower disinfectant concentrations enabling ..."
Par. 3 line 4. Wording - "...which may result in acquired resistance." This seems to imply an increase in acquired resistance mechanisms?	"..., which may result in the acquisition of new resistance mechanisms."
Paragraph 3 lines 7-8. Please place references before the period so the reader knows what statement they are referring to this is inconsistently formatted in other statements in the article.	Sometimes several statements are made within one sentence, each requiring a different reference. In such case we place the references directly after the statements rather than at the end of the sentence.
Par. 3 line 7 "...selection of less susceptible bacteria	"Thus, due to sub-MIC concentrations of

may occur even in the environment." Clarify that environments have low or sub-inhibitory levels of disinfectants. It would be good to include a statement regarding worldwide disinfectant pollution to strengthen this argument.	disinfectants, selection of less susceptible bacteria may occur even in the environment." In Par. 2 we already stated that disinfectants end up in surface waters or soil via sewer lines and fertilizing manure, referring to Tezel 2015.
Par. 3 line 9. "Moreover, resistant bacteria..." what resistance is being referred to here? Disinfectant or antimicrobial or both? It's unclear as written.	We refer to antimicrobials in general, as stated in the previous sentence.
Par. 4 line 2. "...observed in practice offers little reassurance." Who are the stakeholders referred to in this statement?	"That microbial resistance to in-use concentrations of disinfectants has only sporadically been observed in practice offers us little reassurance."
Par. 4 line 4-5. "...routes of incriminated microorganisms,..." what is meant by this term? Is contaminating meant here?	"... transmission routes of micro-organisms that have caused the incident ..."
Par. 5 lines 2-3. "Disinfectants are required to do their work within minutes." Technically disinfectant mechanisms of action also require minute timeframes, perhaps this can be stated?.	What we mean to say is that a comparison between MIC values and in-use concentrations is hampered by differences in timeframes: MIC values are determined in the laboratory, typically after 16-20 hours of exposure, whereas in practice, disinfectants have to do the job in minutes.
Par. 5 line 4. "...application is often not optimal." Please give brief examples of how disinfectant contact is not optimal.	"Contact between the disinfectant and microorganisms at the site of application is often not optimal. Bacteria may reside in places that are difficult to reach by the disinfectant. Also, volatile disinfectants may dissipate too rapidly, while others may be inactivated by organic material."
Par. 5 line 6. Technically bacteria do not "hide in dirt" (this is an anthropomorphism), but the authors argue that the presence of obscuring materials like dirt would impose more barriers and dilute the disinfectant as it reaches and acts on bacteria.	"... (residing in dirt, crevices or biofilms, reducing susceptibilities, enhancing repair mechanisms) ..."
Par. 6 line 2. "Antimicrobial resistance genes tend to be genetically linked..." What is meant by this statement? This is not necessarily true and quite context dependent (eg. Are the linked genes part of an intrinsic resistance vs acquired resistance system, and are linked genes controlled by antimicrobial triggered regulons such as mar-sox-rob system?). It is important to clarify the meaning here to understand what is meant by co-selection and in what form (on plasmids, mobile elements chromosomes)?.	"Antimicrobial resistance genes tend to be genetically linked, by co-residing on plasmids or integrative and conjugative elements, and thus transferred together, paving the way for co-selection."
Par. 6 lines 5-6. "It may also promote HGT of antibiotic resistance genes." Unclear antecedent (what is "it") and sentence fragment.	"It" is referring to "Exposure to disinfectants", the subject of the previous sentence.
Par. 6 line 11. "..., the need for better data is evident." Is better data meant here or more experimental analyses of disinfectant resistance and its associated mechanisms?	"... the need for more data is evident."
Par. 7 lines 1-2. "...,we advocate a prudent approach to all antimicrobials." What is meant by a prudent approach?	"In professional sectors, they should be applied according to evidence-based guidelines specifying when their use has a proven added value in preventing or controlling infection or damage, e.g. food spoilage. Private individuals should only use chemical disinfectants when prescribed by a medical doctor or other qualified experts. In line with international recommendations ²³ health, cosmetic and aesthetic objectives should be pursued without the use of chemical disinfectants whenever possible. In many cases, regular and thorough cleaning with water and a detergent may suffice."
Par. 7 lines 2-3. Can the authors cite supporting literature to support the statement made regarding microbiome toxicity and disinfectant exposure?	We deleted this statement as we agree with the editor that the references to toxicity and interference with the microbiome are out of place here.
Boxed section: What is meant by "redoubling" in line 10 for the statement "After de-doubling, this resulted in 138...".	"After removing duplicate articles, this resulted in 138 publications ..."
Boxed section bullet point 4: "The pathogen was tested for susceptibility to the disinfectant in 13 publications (see Table 1). In most cases, the	"In 12 out of 13 cases, the pathogen was found to be highly resistant to the disinfectant used." According to the EUCAST terminology

pathogen was found to be highly resistant to the disinfectant used." Provide the total number of articles where the pathogen was found to be resistant to the disinfectant. Again how are the authors defining disinfectant resistance (2-fold MIC, 4 fold MIC changes >4 fold MIC) when reviewing these studies?	(microbiological sense)(see ref. 4 of our article), resistance may be low, moderate or high, but EUCAST is not more specific on that. We regard increases in MIC up to 10 times as moderate (see paragraph 1. In most of the 13 studies, the MIC of the pathogen causing the outbreak was not determined. In one study it was determined and was 65 times higher than the MIC of a control strain. In 11 out of 12 other cases we inferred from the information provided by the investigators, and summarised in columns 3, 4 and 5 of our table, that resistance was high or at least high enough to be a co-determinant of the outbreak.
Boxed section summary lines 2-3: "When such a determination was performed, the pathogen was found to be highly resistant to the disinfectant in question in most cases." In most cases undermines that statements made prior. Please give a value.	"In 12 out of 13 cases, the pathogen was found to be highly resistant to the disinfectant used." (see above)
Box section conclusion line 1: "Resistance to disinfectants likely does play an important role" please remove the word "does" and pluralize "play".	"So, resistance to disinfectants likely plays an important role in incidents involving disinfection failure in daily practice ..."
Table 1 Layout	We are glad to adapt to the requirements of the journal.
Reviewer 2	
I strongly suggest to use the current terminology used in the European Union. Disinfectants are biocidal products based on different biocidal active substances. This is important because disinfectants may be based on two or more different biocidal active substances. Increased MIC values indicating cellular tolerance are typically determined using a single biocidal active substance although it may be possible to determine an MIC value using the entire formulation. This aspect should be considered in the entire manuscript.	"Disinfectants are antimicrobial products that incorporate one or more active substances such as chlorine, iodine, alcohols, hydrogen peroxide, silver, chlorhexidine, triclosan and quaternary ammonium compounds." We agree that MICs can be determined of single active substances, of mixtures of active substances and of entire formulations, but we think that information is too much detail for this paper and less relevant for our message.
2. I agree that there is a lack of international definition to determine "resistance". But I am also not clear about the use of the terms "tolerance" and "resistance" in the manuscript. My understanding is that tolerance describes any type of elevated MIC or MBC values compared to comparable isolates or strains of a species. The term "resistance" may indicate an epidemiological resistance as suggested by Morrisey et al. (10.1371/journal.pone.0086669) or a clinical resistance describing an isolate with a lower log reduction compared to type of recommended application (10.1016/s1473-3099(03)00833-8). This aspect should also be considered in the entire manuscript.	"In line with EUCAST ⁴ , we advocate to use the term 'tolerance' only in cases where the MBC of a strain is much increased (the strain is not readily killed anymore), while its MIC remains unchanged. In microbiological sense, the term 'resistance' is used to denote any reduction in susceptibility demonstrated phenotypically by increases in MIC or MBC."
3. "sub-lethal concentrations": please include sub-inhibitory concentration because they have been studied in numerous publications with an adaptive cellular response.	"Exposure to sub-MIC concentrations of disinfectants trigger stress responses in bacteria that induce temporary, adaptive changes in ..."
4. "Exposure to very low concentrations": please be more specific (see 3.).	"... that exposure to concentrations of antimicrobials far below the MIC enable bacteria ..."
5. Page 3, par.3: please consider to add volatility of an important factor for a possible adaptation (duration of exposure) and stability of the chemical (autocatalytic substances; also duration of exposure and concentration during exposure).	"Contact between the disinfectant and microorganisms at the site of application is often not optimal. Bacteria may reside in places that are difficult to reach by the disinfectant. Also, volatile disinfectants may dissipate too rapidly, while others may be inactivated by organic material."
6. Page 3, last paragraph: cross-tolerance to other biocidal active agents also occurs and would be worth to mention with one or two examples.	"Mechanisms that reduce a microorganism's susceptibility to a disinfectant, may also diminish its susceptibility to other disinfectants and antibiotics, a phenomenon called cross-resistance. ¹⁸ "

REVIEWERS' COMMENTS:

Reviewer #1 (Remarks to the Author):

The authors have satisfactorily addressed all the major concerns regarding the manuscript. There are a few grammatical issues in some of the revised sections but these can likely be addressed at a proofing stage. There are no additional comments requested.

Reviewer #2 (Remarks to the Author):

Some of my concerns have been addressed while for the other concerns the authors have described a reason why they did not follow them. Overall the manuscript may now be acceptable as a comment because the topic is of increasing relevance. One final remark is left. Lines 74-78: It is described as "disinfectants trigger", I would have preferred to read "disinfectants can trigger" or "disinfectants may trigger". In addition, this potential for an adaptive response has been described only for some biocidal agents, especially triclosan CHG and BAC. Other biocidal agents have been investigated but an adaptive response was not found. In its current form the sentences are too general. I suggest to change the wording.

Response to referees 2

Point raised by reviewer 2	Response of authors
"disinfectants can trigger" or "disinfectants may trigger"	"Exposure to sub-MIC concentrations of disinfectants can trigger stress responses in bacteria ..."